# Development of Coated Electrodes with Solid Wire and Flux-Cored Alloyed Wire for Microalloyed Steel Welding

**DOI:** 10.3390/ma13092152

**Published:** 2020-05-06

**Authors:** Darko Bajić, Mihailo Mrdak, Nikola Bajić, Darko Veljić, Marko Rakin, Zoran Radosavljević

**Affiliations:** 1Faculty of Mechanical Engineering, University of Montenegro, Bul. Džordža Vašingtona bb, 81000 Podgorica, Montenegro; 2Innovation Center, Faculty of Technology and Metallurgy, University of Belgrade, Karnegijeva 4, 11000 Belgrade, Serbia; drmrdakmihailo@gmail.com (M.M.); veljic.darko@gmail.com (D.V.); 3IHIS Techno-experts-Research and Development Center, Batajnički Drum 23, 11080 Belgrade, Serbia; nikola.bajic49@gmail.com; 4Department of General Technical Sciencies, Faculty of Technology and Metallurgy, University of Belgrade, Karnegijeva 4, 11000 Belgrade, Serbia; marko@tmf.bg.ac.rs; 5Research and Development institute Lola L.T.D., 70A Kneza Viseslava Street, 11030 Belgrade, Serbia; rzoran67@yahoo.com

**Keywords:** weld metal, arc welding, mechanical properties, impact toughness, microstructure

## Abstract

In this paper, we will present our investigation of the quality of J55 microalloyed steel welds that were formed by a basic flux-cored wire electrodes that were of appropriate quality and alloyed with Ni and Mo. Based on the comparison and analysis of the obtained results related to the testing of the chemical composition, mechanical properties, toughness at test temperatures, and the microstructure of welding joints formed by a classic and specially coated rutile flux-cored electrode, we assessed the justification to switch from solid wire electrodes to flux-cored alloyed wire electrodes of appropriate quality. The research aim for the application of flux-cored wire electrodes instead of solid wire electrodes is based on the advantages pertaining to a flux-cored wire: molten metal from electrode wire is transferred in the form of fine droplets, easy welding and maximum productivity within all spatial positions related to welding, improved properties of welding joints, and increased productivity when compared to a classic solid wire. Our research encompasses the development of the experimental production at the Research and Development Center IHIS Belgrade (Development Institute for Chemical Power Sources), Serbia, of the new type of a coated electrode with improved welding properties when compared to a classic electrode intended for microalloyed steel welding.

## 1. Introduction

The development and production of a coated electrode of improved welding properties are a part of a complex research project, which necessitates a change of certain technological phases related to the manufacturing of electrode core and coating [1,2,3]. It is possible to improve electrode properties if we replace solid wire with a flux-cored alloyed wire. If we want to use solid wire as an electrode core, we need a sufficient thickness of the steel sheath so we can make an electrode of a required diameter and length with flat ends, which can be coated evenly by applying continuous pressure. The goal of the paper was to master the production of new coated rutile electrodes with a core of flux-cored wire with improved quality compared to conventional electrodes. Emphasis was placed on developing coated rutile electrodes with flux-cored wires that allow the formation of weld metal (WM) with a planned chemical composition and desired mechanical properties. For experimental welding, we selected steel plates made from microalloyed steel, mark J55 (according API Spec 5L) of standards (EN 10113–3. and JUS C. B0 502.), produced in Smederevo Steelworks (Smederevo, Serbia). The results of testing the mechanical properties and the microstructure of the WM of welded joints made using specially coated rutile electrodes with cores of flux-cored wire, produced in IHIS Belgrade (Development Institute for Chemical Power Sources), confirmed that the electrodes are of reliable quality.

The basic criteria that have to be taken into account when designing and manufacturing filler materials are yield strength, tensile strength, and weld metal (WM) toughness. When designing the quality of coated electrodes used for the welding of microalloyed steel of increased strength, which is susceptible to cracking within the welding joint zone, it is necessary that filler materials contain elements that increase crack resistance, as well as the elements that enhance the WM properties. Apart from filler materials, WM also contains impurities that all together affect the properties of a welding joint. Elements that are added with the aim of alloying are Mn, Cr, Ni, and Mo, while the elements such as Si, Ti, Al and Zr are added in order to remove or limit the amount of O and N, while S and P are deemed to be impurities [4]. Base electrode coating, together with a flux-cored alloyed wire, has a multifunctional purpose because they enable the stabilization of the electric arc and, at the same time, additional alloying, smooth slag formation, and the degassing of molten metal during a welding process [5,6]. The combination of coating and flux-cored wire necessitates a precise definition of the composition of the coating and the composition of the flux-cored wire, since only a limited amount of oxygen may be present in the weld metal of low-alloyed steels, which is responsible for the formation of a high percentage of acicular ferrite within the WM structure.

A flux-cored wire consists of a metal sheath and a core. The metal sheath is made of a thin low-carbon low-alloy steel sheet whose quality is very close to the quality of the material used to make solid wires. Flux-cored wires contain the materials that are composition-wise very similar to the materials used to make the coating. The core also contains metal components used for alloying the weld metal (Mn, Ni, Mo, and Cr, etc.), and, if necessary, metallic and non-metallic components which, together with the coating, are aimed at creating slag (TiO_2_, Si, Zr), stabilizing the electric arc (feldspar, mica, potassium oxalate), creating gases (CaCO_3_, MgCO_3_, etc.), deoxidation (Si, Mn, Fe, Ti, Al-powder), and increasing the effectiveness of the melting and welding speed (iron powder).

Basic coated electrodes chosen for the welding of microalloyed steels are experimentally manufactured at the Research and Development Center IHIS Belgrade, Serbia. In order to test the impact of alloying elements Ni, Mn, and Mo on the quality of welding joints, the center has experimentally manufactured two solid wire electrodes and two flux-cored wire electrodes. When it comes to the solid wire electrodes, the content of Ni has two values (1.10% Ni and 2.5% Ni), while the content of Mo and Mn is kept at the same level. With this type of electrode, the alloying elements are inserted into WM via the electrode coating. On the other side, when it comes to the flux-cored wire electrodes, the content of Ni has two values (1.10% Ni and 3.3% Ni), as well as Mn (0.90% Mn and 1.15% Mn), while the content of Mo is at the same level. With this type of electrode, the alloying elements are inserted into WM via the flux-cored wire.

The obtained results of the research represent a foundation for a reliable definition of welding regime parameters and the selection of an appropriate type of filler materials, together with ensuring the quality of microalloyed steel welding joints. If we increase the quality of welding joints, that represents a significant contribution regarding the reliability of these steels when they are used for the production of large-diameter seamed pipes, major pipelines, and pressure vessels.

Investigation of the welding joint structure encompasses the use of a microscope to observe structural changes within the heat-affected zone (HAZ) and WM. Quality analysis of the structure and properties of the HAZ is a precondition for the assessment of the structural safety of welding joints. What kind of a welding joint we will get depends on the properties of base metal (BM), the thermal regime of welding, and the cooling time within the temperature range from 800 to 500 °C.

HAZ width depends on the welding procedure, i.e., on its parameters. If we change those parameters, we change the welding heat input. Structural changes within a HAZ segment correspond to the temperature level of such a segment: molten weld pool, partial melting, overheating, normalizing, precrystallization, and recrystallization.

The basic criterion that has to be met when selecting filler materials is the value of yield strength, tensile strength, and WM toughness. Considering the fact that specific types of microalloyed steels of higher strength are more susceptible to cracking, it is necessary that filler materials contain elements that increase the crack resistance, as well as the elements that enhance the WM properties. Apart from filler materials, which are added deliberately, the WM also contains impurities which all together affect the properties of a welding joint. Elements that are added with the aim of alloying are Mn, Cr, Ni, and Mo, while the elements such as Si, Ti, Al, and Zr are added in order to remove or limit the amount of oxygen and nitrogen, while S, P, and O are deemed to be impurities [7,8,9].

The quality of filler materials and welding regime are significant factors affecting the welding joint quality. Cooling speed is directly linked to welding parameters, and this speed is defined by the amount of heat input while welding. The two most important properties of a welding joint are WM and HAZ, as the components of such a welding joint. Due to this, during welding, it is necessary to achieve the most favorable microstructure of HAZ and WM as a precondition for the achievement of optimal mechanical properties of a welding joint and structural safety in terms of its exploitation [10,11]. The most favorable microstructure and mechanical properties has a WM with a high proportion of acicular ferrite (AF), whose proportion directly depends on the chemical composition of the WM and its cooling speed [12]. The chemical composition of WM is, mainly, defined by the type of added filler materials and, to a lesser degree, by BM, which melts during welding. In general, the impact of alloying and microalloying elements on the microstructure of WM, through individual effects and interactions, is as follows. Carbon in austenite is always dissolved to the level of atoms. It is known that austenite-to-ferrite and austenite-to-perlite transformation are completed by the diffusion of carbon, while transformations to bainite and martensite take place without carbon diffusion. If WM contains a reduced amount of carbon, the hardenability reduces proportionally (we cannot obtain the needle-like microstructure of martensite). Furthermore, a lower amount of carbon has a favorable impact on WM after welding, thanks to the improvement in microstructure. The most favorable structure within a ferrite seam is obtained if the carbon content is around 0.06%. Manganese is essential for the kinetics of phase transformation in welds, which consists in the limitation of growth of primary ferrite up to the moment when the degree of undercooling reaches the level where the creation of acicular ferrite is energetically possible. The high content of Mn within a weld seam restricts the formation of acicular ferrite, which is probably due to the further decomposition of primary austenite grains within WM. If we introduce manganese to WM, that will reduce the negative impact of sulfur by binding it to an appropriate sulfide. In terms of low-alloy steels and their welding joints, it is considered that the optimum amount of Mn within WM is about 1.5%. The content of Mn above 1.6% is conducive to the formation of upper bainite, while this decreases the content of needle-like ferrite. Silicon increases the growth rate of primary ferrite and promotes the formation of Widmanstätten ferrite, thus reducing the possibility of the creation of acicular ferrite. Additionally, silicon reduces the thermodynamic activity of iron and hinders the creation of the carbide phase by displacing carbon from the iron solution. This is conductive to the significant enrichment of austenite by carbon. The significant impact of Si on the hardenability within a solid solution has a detrimental effect on WM, so it is desirable to keep its content below 0.4%. Nickle is conducive to the formation of needle-like ferrite. Nickel, within a solid solution, increases the resistance of WM to brittle fracture, and that is the reason it is a common alloying element of WM of fine-grained microalloyed steels of increased strength [7,8,9,12]. Molybdenum is considered very important in terms of reaching optimum ratios between different structural components. The addition of molybdenum decreases the transition temperature (α → γ), increases the hardenability, and reduces the size of the WM’s primary structure. Then, this increases the quantity of acicular ferrite and almost completely removes upper bainite, keeping only thin platelets of primary ferrite within the matrix of WM [7,8,9]. Titanium, if introduced in a very small quantity into a filler material, completely modifies the WM microstructure. Titanium has a significant impact on toughness. Aluminium with 35 ppm of titanium in WM impacts the change in transition temperature. We may state that the impact of this content of titanium, coupled with approximately 400 ppm of aluminium, is insignificant in terms of welding. Niobium, when added to WM, increases the size of ferrite grain (10 μm), and the addition of Nb and Ti decreases the size of ferrite grain (3 μm). At the same time, with the introduction of Nb and Ti, ferrite hardness increases (10–15%). Vanadium, when present in WM and C-Mn steels, may be tolerated up to 200 ppm [7,8,9,12].

## 2. Materials and Methods

### 2.1. Base Metal and Filler Materials 

For the selected base metal for the experimental welding, we have chosen J55 microalloyed steel according to API Spec 5L standard (EN 10113–3. and JUS C.B0 502), thickness: 7.0 mm, manufactured in Smederevo steel mill, Serbia. The designation, chemical composition, and mechanical properties in terms of rolling of API 5CT J55 microalloyed steel are shown in Table 1, as well as the values according to the standard [13].

For the experimental welding, we have chosen two types of electrodes with solid wire and two types of electrodes with flux-cored wire aimed at welding microalloyed steel. In order to test the impact of the alloying elements Ni, Mn, and Mo on the quality of welding joints, we obtained four types of experimentally manufactured electrodes in which the alloying elements vary at two levels. The first two experimentally manufactured types of electrodes with solid wire have internal designations: IHIS 1.1Ni Mo B and IHIS 2.5Ni Mo B. Here, the electrode coating is responsible for the introduction of alloying elements into WM. The other two experimentally manufactured types of electrodes with flux-cored wire have the following internal designations: IHIS 1.1Ni Mo B-pp and IHIS 3.3Ni Mo B-pp. With this type of electrode, flux-cored wire is responsible for the introduction of alloying elements into WM. Flux-cored wire of necessary quality and the selected types of coated electrodes were experimentally made in the laboratory of the Development and Research Centre, IHIS Belgrade, Serbia. Flux-cored wire, whose diameter is Ø3.25 mm, was made of narrow steel wire, 10 mm wide and 0.8 mm thick, with the following designation of steel quality: Č.0147 (JUS C.B4.016), made in Smederevo steel mill, Serbia. The designations and chemical composition of experimentally manufactured coated electrodes (according to standard: EN499/757, E5522NiMoB42) are given in Table 2.

### 2.2. Experimental Welding

For the experimental welding, we cut several 150 × 250 mm sheets from 7 mm thick J55 microalloyed steel, whose edges were then prepared according to Figure 1.

Experimentally manufactured coated electrodes of internal designation IHIS 1.1.Ni Mo B, IHIS 2.5Ni Mo B, IHIS 1.1Ni Mo B-pp and IHIS 3.3Ni Mo B-pp, before welding, had been dried at the temperature of 250 °C for 3 h.

Experimental welding of the prepared samples (sheets) made of J55 steel was performed using manual arc welding (MMAW) according to the parameters shown in Table 3. Welding of the samples was completed using a selection of filler material and a low level of heat input from 7 to 7.7 kJ/cm.

## 3. Results and Discussion

### 3.1. Testing of the Chemical Composition of Pure WM

Table 4 shows the chemical composition of pure WM of welded samples with four different qualities of coated electrodes, out of which two were with solid wire and the other two with flux-cored wire.

### 3.2. Testing of Mechanical Properties of Welding Joints

Test results of the tensile strength and bending angles of experimentally welded samples of J55 microalloyed steels with an appropriate quality of filler material are shown in Table 5. Lower values of bending angle α at weld root are due to the impact of non-metallic inclusions, which are present in the weld root.

### 3.3. The Charpy Impact Test

A standard Charpy test was used to test the toughness of experimentally welded samples of J55 using a manual arc welding. Figure 2 shows the results of testing the toughness of J55 hot rolled microalloyed steel (batch no. 250046, Smederevo steel mill, Serbia).

Figure 3, Figure 4, Figure 5 and Figure 6 show the obtained results for toughness, at test temperatures, of welding joints within the HAZ and WM zone. The diagrams (Figure 3 and Figure 4) show the changes in toughness depending on the test temperature for the WM and HAZ of J55 base metal (BM) when using solid wire electrodes with the following designations: No.1.1 (0.95% Mn + 1.04% Ni + 0.31% Mo) and No.1.2 (1.34% Mn + 2.850% Ni + 0.59% Mo). Figure 3 and Figure 4 show a small increase in toughness due to the increased content of Ni and Mn in WM.

The diagrams (Figure 5 and Figure 6) show the changes in toughness depending on the test temperature of WM and HAZ of J55 BMl when using flux-cored electrodes with the following designations: No.2.1 (0.68% Mn + 1.868% Ni + 0.241% Mo) and No.2.2 (0.89% Mn + 3.245% Ni + 0.307% Mo). Figure 5 and Figure 6 show a small increase in the toughness of HAZ and WM due to the increased content of Ni and Mn in WM.

### 3.4. Fractographic Analysis of WM Toughness

By using a scanning electron microscope (SEM), we have performed a fractographic analysis of samples at the place where the fracture had occurred, while testing the toughness of WM of welding joints made of J55 microalloyed steel. The weld was made using the following electrodes: IHIS 1.1Ni Mo B (sample No.1.1) and IHIS 2.5Ni Mo B (sample No.1.2) after the fracture had occurred at the following temperatures: +20, −40, and −60 °C.

Fractures within WM at −40 °C are ductile in their nature. Measured values are according to the microstructure, which was analyzed by SEM. Impact energy values for all samples show that the WM did not contain local brittle zones (LBZ). A fine-grained structure of acicular ferrite led to an increased dissipation of impact energy and improved toughness. The highest value of toughness, 75 J, was achieved in WM with the highest content of Ni and Mo. An increased amount of Ni had a favorable impact on toughness. Nickel was conductive to the formation of needle-like ferrite in WM, which increased its resistance to brittle fracture. Molybdenum additionally reduced the size of the primary structure of WM, reduced the content of proeutectoid ferrite, increased the amount of acicular ferrite, and completely removed upper bainite [9]. The WM fracture properties show that the test weld of microalloyed steel is utterly reliable at −40 °C.

Figure 7 show the fracture surface view of WM samples (Nb/Ti J55) tested for their toughness. WM was created using the electrode IHIS 1.1Ni Mo B (sample No.1.1), which was alloyed with (0.95% Mn + 1.04% Ni + 0.31% Mo), and this WM fractured at the following temperatures: +20, −40, and −60 °C.

The WM sample, created by using the electrode No.1.1 at +20 °C, was ductile along the whole cross-section (Figure 7a). We can see the elongation of pits in the direction of deformation and the intermittent presence of carbides and other particles in the pits.

The mixed-mode fracture of WM, which is in the terms of its depth relatively even, was obtained while testing at the temperature of −40 °C. Photographs of the microstructure (Figure 7b) show the look of a mixed-mode fracture where the dominant fracture is transcrystalline brittle fracture, which is caused by the deposit. Figure 7c shows the microstructure of the fracture surface height-wise. The shown sample fractured at −60 °C. This photograph of the microstructure gives the general view of brittle fracture caused by tearing in the middle portion of the seam. However, this sample also shows micro zones with pit-like fractures.

Figure 8 show the fracture surface of WM created by the electrode IHIS 2.5Ni Mo B, No.1.2 (1.34% Mn + 2.850% Ni + 0.59% Mo) at the following temperatures: +20, −40, and −60 °C.

Photography of the microstructure (Figure 8a) shows the characteristic fracture surface of WM, which was created by the same electrode, but at even a lower test temperature of −60 °C. In this case, this is a completely brittle fracture of transcrystalline and intercrystalline character.

### 3.5. Base Metal Microstructure

Figure 9 shows the microstructure of J55 microalloyed steel, which was analyzed under the optical microscopy (OM) (**a**,**b**) and the SEM (**c**,**d**) microscope.

### 3.6. Microstructural Analysis of WM

Microstructural analysis of the WM of all samples shows a significant amount of acicular ferrite (AF). The volume of acicular ferrite within the WM of J55 microalloyed steel, which was created by the following electrode—IHIS 1.1Ni Mo B (sample No.1.1) with (0.95% Mn + 1.04% Ni + 0.31% Mo)—was between 50% and 72.85%, while the other forms of ferrite were proeutectoid ferrite (PF) and ferrite with secondary phase (FS). The SEM analysis of WM confirmed that the primary microstructure of WM contains three types of ferrite: acicular ferrite (AF), proeutectoid ferrite (PF), and ferrite with secondary phase (FS). Figure 10 shows the OM (**a**) and SEM (**b**–**d**) microstructure of the WM created by the sample No.1.1 electrode. This microstructure clearly shows a narrow longitudinal line of proeutectoid ferrite (PF), which was separated at the border between austenite grain and needles of acicular ferrite (AF).

Austenite grains show spherical non-metallic inclusions from the basic coating, which serve as sites for the intragranular nucleation of needle-like ferrite [8,12,13,14,15,16,17]. The volume of acicular ferrite (AF) within the MW of J55 microalloyed steel created by the electrode IHIS 2.5Ni Mo B (sample No.1.2) with (1.34% Mn + 2.850% Ni + 0.59% Mo) ranged from 56% to 79.44%, while the other forms of ferrite were proeutectoid ferrite and ferrite with secondary phase. Figure 11 shows the OM (**a**,**b**) and SEM (**c**,**d**) analysis of the microstructure of WM created by the sample No.1.2 electrode. This microstructure consists of austenite grains with separated proeutectoid ferrite (PF) along the border of the grain. Metallographic analysis of WM shows that the increase in Ni increases the content of acicular ferrite (AF) at the expense of proeutectoid ferrite (PF). The inside of primary austenite grains is composed of acicular ferrite (AF) characterized by randomly oriented needle-like crystals arranged in a net-like pattern.

This natural intertwining of ferrite needles, coupled with its fine grains, provides good toughness and resistance to factures. A small amount of molybdenum in WM, 0.30%, resulted in finer primary structure and, coupled with Ni, it increased the amount of acicular ferrite. The WM microstructure did not contain the secondary phases of bainite and perlite, which are formed as a result of thermal treatment caused by every new weld pass.

The WM of J55 microalloyed steel created by the electrode IHIS 1.1Ni Mo B-pp (sample No.2.1) with (0.68% Mn + 1.868% Ni + 0.241% Mo) ranged from 56% to 79.44%. Figure 12 shows the OM (**a**,**b**) and SEM (**c**,**d**) analysis of the microstructure of WM created by the sample No.2.1 electrode. The highest volume of acicular ferrite, in the last weld pass, was present in the WM of J55 microalloyed steel created by electrode IHIS 3.3Ni Mo B-pp, which is alloyed with the highest content of Ni, sample No.2.2 (0.89% Mn + 3.245% Ni + 0.307% Mo). Figure 13 shows the OM (**a**,**b**) and SEM (**c**,**d**) microstructure of the WM alloyed with the highest amount of Ni, which was 3.238%. The WM microstructure is mainly composed of acicular ferrite (AF), which ranged from 58% to 87.3%, with smaller amounts of proeutectoid ferrite (PF) and ferrite with secondary phase (FS). The acicular grains of ferrite grow from small particles of white inclusions, which are evenly distributed within austenite grains. The present non-metallic inclusions are mostly oxides and are predominantly spherical.

## 4. Conclusions

Based on the theoretical analysis and the results obtained through our experimental investigation of the impact of the quality of experimentally manufactured coated electrodes with solid wire and flux-cored wire on the properties of welding joints of J55 microalloyed steel according to API Spec 5L standard (EN 10113–3), we can draw the following conclusions. Mechanical–technological properties of WM are primarily characterized by the chemical composition of filler material, i.e., the amount of inserted alloying elements. The total amount of alloying elements inserted into the weld metal from the coated electrode No.1.1 amounted to 2.92%; No.1.2 amounted to 5.66%; No.2.1 amounted to 2.99%; and No.2.2 amounted to 5.13%. The above-mentioned percentages of alloying elements inserted during the technological process of weld metal formation hints at their possible impact on the quality of weld metal, taking into consideration the percentage of their mixture with base metal.Testing of weld metal toughness shows that the highest impact on the toughness can be attributed to the chemical composition of base metal and the reaction of elements during the welding procedure. The highest values for toughness were achieved with weld metals created by the No.1.2 and No.2.2 electrodes because, apart from the existence of Ni and Mo in weld metal, this led to the increase in Mn from base metal caused by mixing within the molten weld pool and the reduction in carbon due to its burn-out, which has a favorable impact on the formation of acicular ferrite within WM. It can be concluded that toughness, at lower temperatures, is higher than 25–40%.Fractographic analysis of fracture toughness of J55 weld metal created by No.1.1 and No.2.1 electrodes at +20 °C shows that the fracture is ductile along the whole cross-section. Weld metal brittle fracture increases in frequency with specimens which were fractured at −40 °C, where the transcrystalline fracture becomes more prevalent. With specimens that were fractured at −60 °C, the brittle fracture of transcrystalline and intercrystalline character become visible.The best quality of welding joints in terms of J55 microalloyed steel of increased strength was achieved by the electrodes No.1.2 (1.34% Mn + 2.850% Ni + 0.59% Mo) and No.2.2 (0.89% Mn + 3.245% Ni + 0.307% Mo), with low heat input (7.5–8.5 kJ/cm).Regardless of the fact that the price of flux-cored electrode is higher than the solid wire electrode, it is still a better option, since it yields better weld metal properties related to microalloyed steel if other filler materials are of sufficient quality.

## Figures and Tables

**Figure 1 materials-13-02152-f001:**
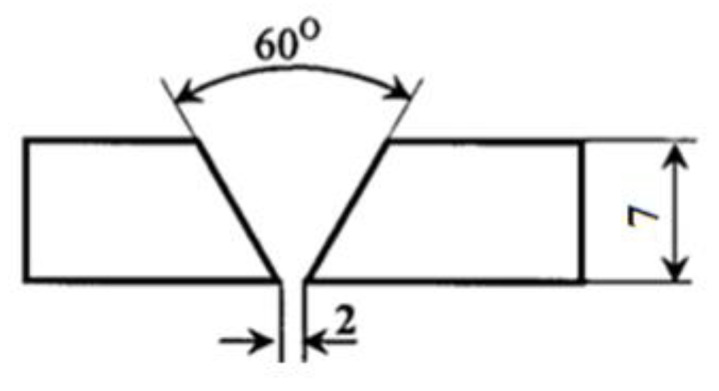
Preparation of the edges of experimental J55 steel samples.

**Figure 2 materials-13-02152-f002:**
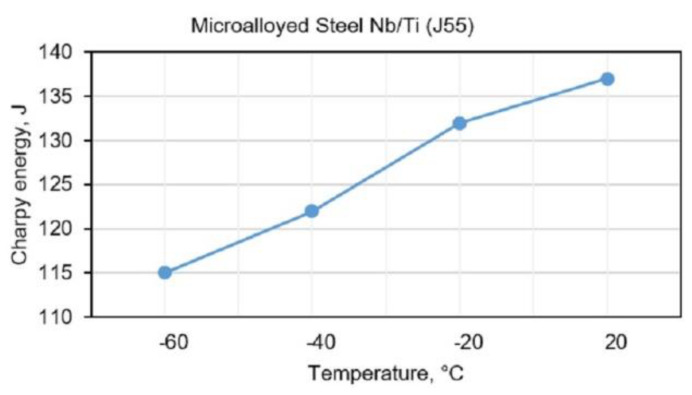
Impact toughness of BM Nb/Ti (J55) at different temperatures.

**Figure 3 materials-13-02152-f003:**
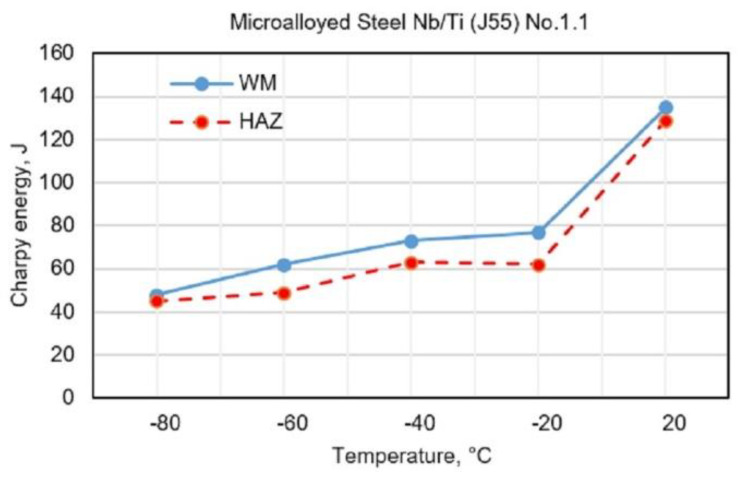
Test results of impact energy at test temperatures of welded joints made with electrode No.1.1.

**Figure 4 materials-13-02152-f004:**
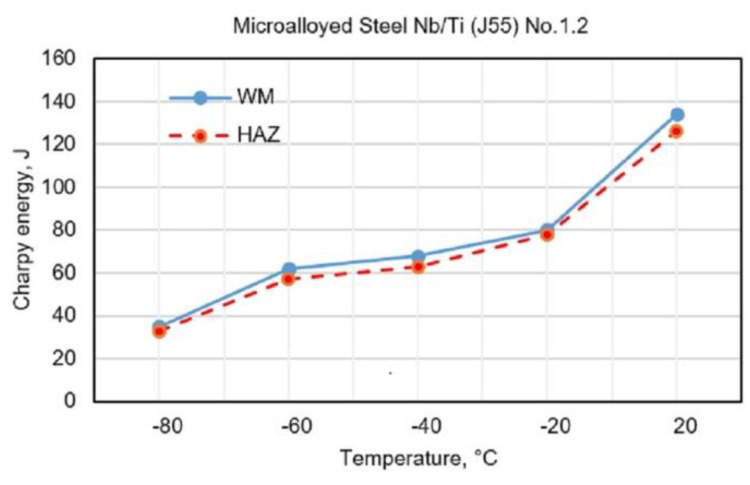
Test results of impact energy at test temperatures of welded joints made with electrode No.1.2.

**Figure 5 materials-13-02152-f005:**
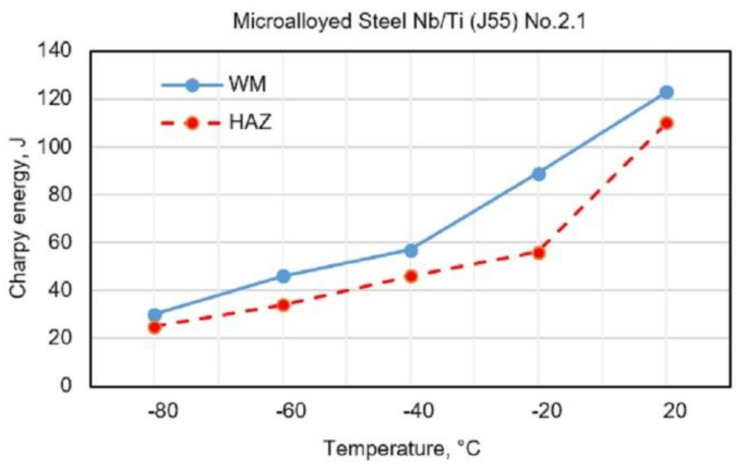
Test results of impact energy at test temperatures of welded joints made with electrode No.2.1.

**Figure 6 materials-13-02152-f006:**
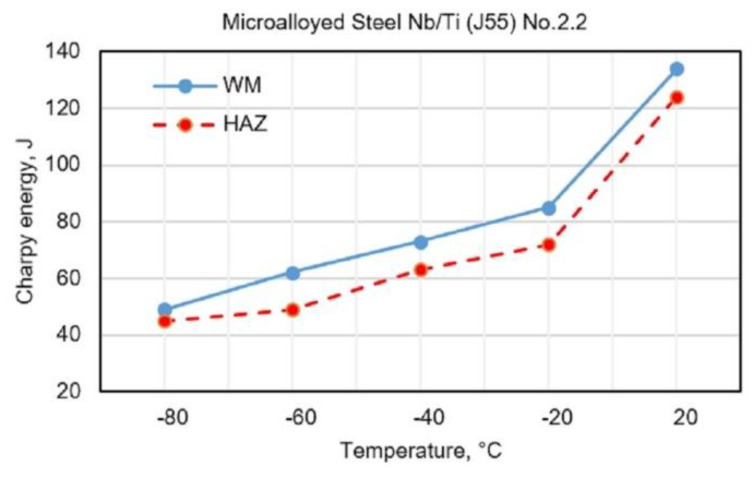
Test results of impact energy at test temperatures of welded joints made with electrode No.2.2.

**Figure 7 materials-13-02152-f007:**
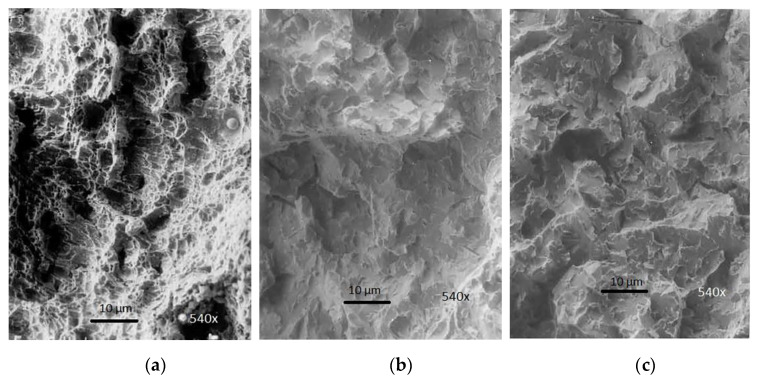
SEM fracture surface view of testing of toughness of WM formed by using electrode No.1.1: (**a**) (*t* = +20 °C), ductile fracture; (**b**) (*t* = −40 °C), brittle fracture of transcrystalline and intercrystalline character in the middle of WM; (**c**) (*t* = −60 °C), brittle fracture of transcrystalline and intercrystalline character in the middle of WM.

**Figure 8 materials-13-02152-f008:**
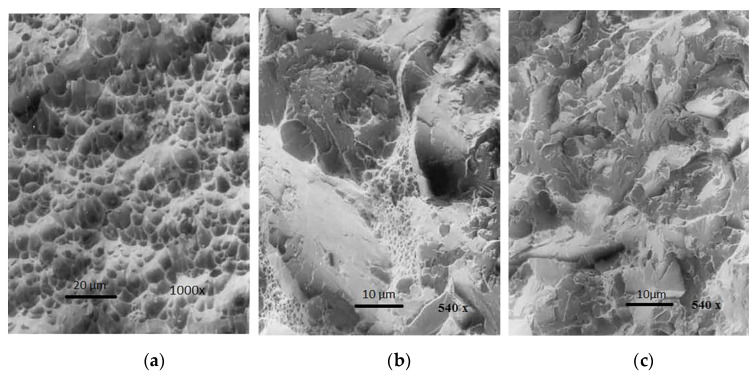
SEM fracture surface view after testing the toughness of WM created by the electrode No.1.2: (**a**) (*t* = + 20°C), ductile fracture with tiny pits containing deposited particles; (**b**) (*t* = −40 °C), mixed-mode fracture, which is predominantly transcrystalline; (**c**) (*t* = −60 °C), transcrystalline brittle fracture with the presence of microvoids in the middle of WM.

**Figure 9 materials-13-02152-f009:**
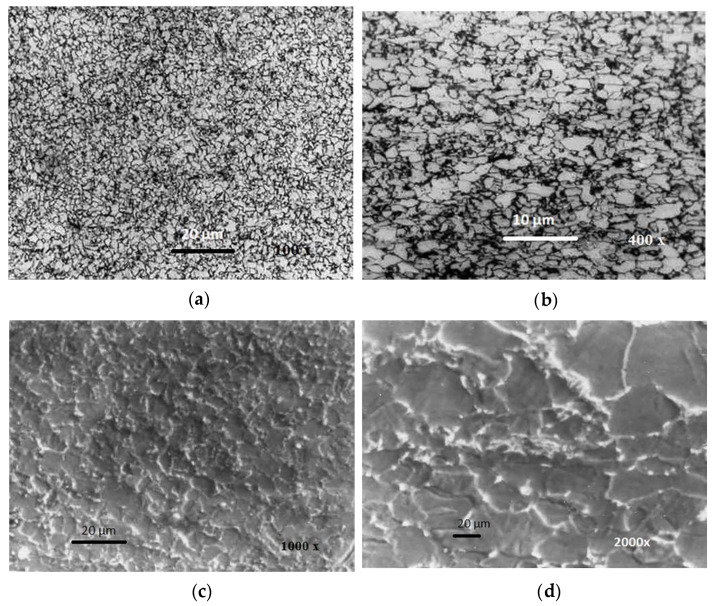
OM (**a**,**b**) and SEM (**c**,**d**) microstructure of the BM API 5CT J55.

**Figure 10 materials-13-02152-f010:**
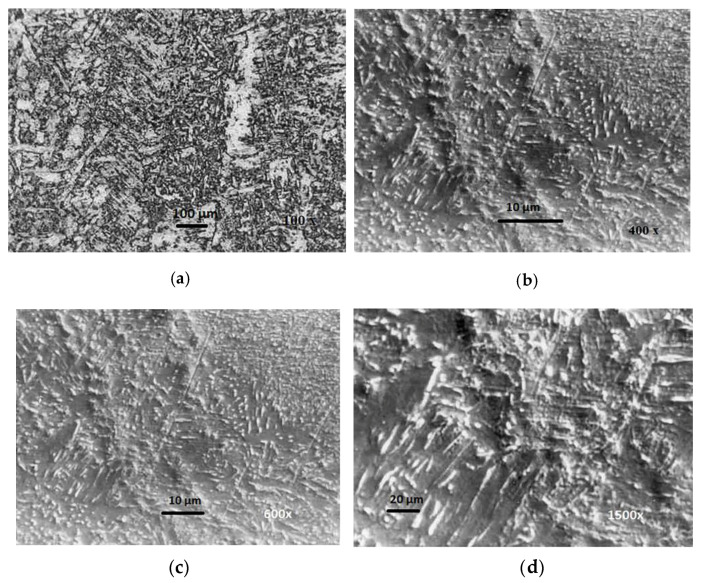
OM (**a**) and SEM (**b**–**d**) microstructure of WM created by electrode No.1.1.

**Figure 11 materials-13-02152-f011:**
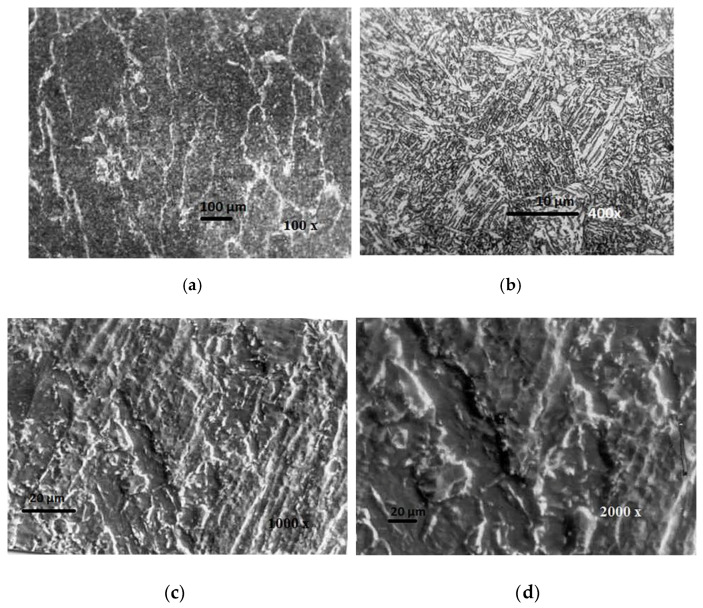
OM (**a**,**b**) and SEM (**c**,**d**) microstructure of WM created by electrode No.1.2.

**Figure 12 materials-13-02152-f012:**
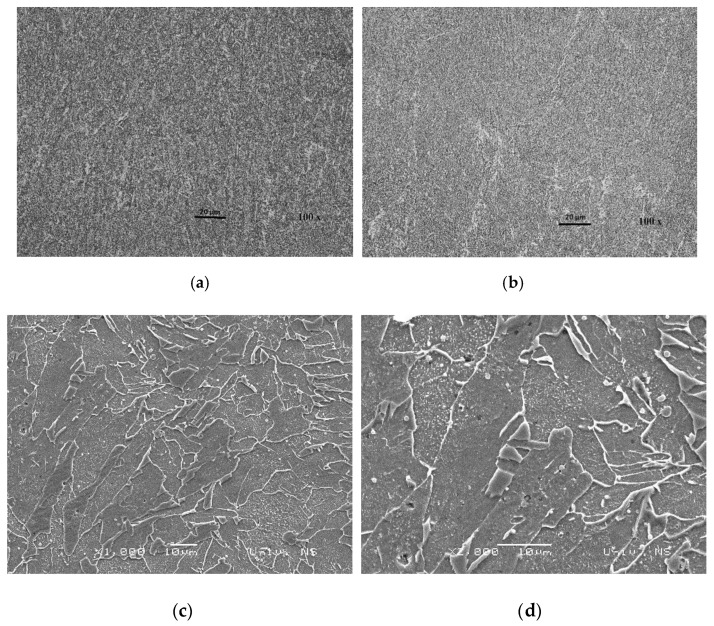
OM (**a**,**b**) and SEM (**c**,**d**) microstructure of WM created by electrode No.2.1.

**Figure 13 materials-13-02152-f013:**
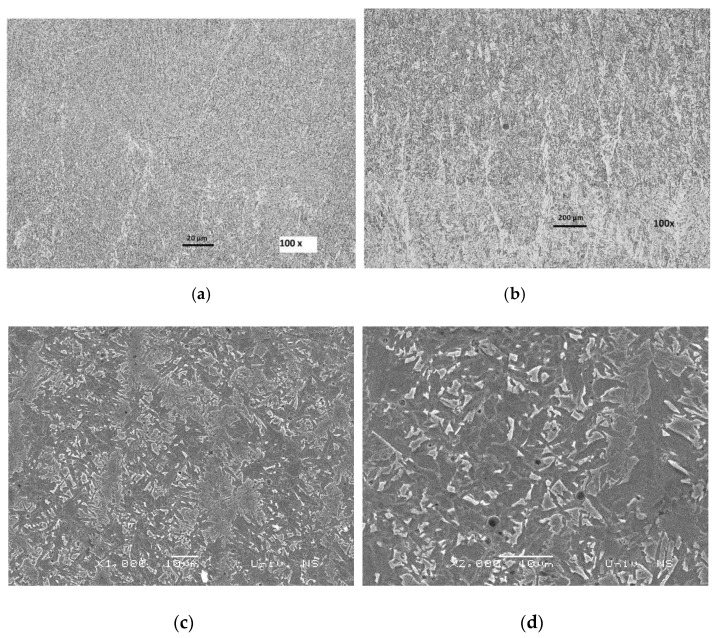
OM (**a**,**b**) and SEM (**c**,**d**) microstructure of WM created by electrode No.2.2.

**Table 1 materials-13-02152-t001:** Chemical composition and mechanical properties of microalloyed steel API 5CT J55 in the direction of rolling.

Steel API 5CT	Chemical Composition, wt.%									Mechanical Properties		
	C	Si	Mn	P	S	Al	Cu	Ti	Nb	*R*_e_, MPa	*R**_m_*, MPa	*A_5_*, %
**J 55**	0.06	0.26	1.18	0.02	0.007	0.031	0.045	0.013	0.035	497	578	32
	API 5CT Standard									379–552	>517	>22.5

**Table 2 materials-13-02152-t002:** Designations of the coated electrodes, their chemical composition, and mechanical properties of weld metal (WM), according to the specifications of the manufacturer.

Designation	Comparative Nomenclature with Other Manufactures	Chemical Composition, wt %				Mechanical Properties of WM				
		C	Si	Mn	Ni	Mo	*R_e_*, MPa	*R_m_*, MPa	*A*_5_, %	KV, J (−40 °C)
IHIS 1.1Ni Mo B	Jesenice, Slovenia: EVB NiMo	0.06	0.40	0.90	1.10	0.35	510	580–710	22	47
IHIS 2.5Ni Mo B	Plužine, Montenegro: PIVA255BMo	0.08	0.5	0.95	2.5	0.35	550–640	650–750	22–26	60–90
IHIS 1.1Ni Mo B-pp	Jesenice, Slovenia: Galeb70	0.06	0.40	0.90	1.1	0.35	510	580–710	22	47
IHIS 3.3Ni Mo B-pp	Jesenice, Slovenia: EVB 2,5NiMo	0.06	0.45	1.15	2.30	0.40	590	650–750	20	47

**Table 3 materials-13-02152-t003:** Welding parameters for microalloyed API 5CT J55 steel samples using the arc process.

Electrode Designation	Electrode Diameter, mm	Welding Current *I*, A	Arc Voltage *U*, V
IHIS 1.1Ni Mo B	3.25–smw	120	25
IHIS 2.5Ni Mo B	3.25–smw	120	25
IHIS 1.1Ni Mo B-pp	3.25–fce	80	25
IHIS 3.3Ni Mo B-pp	3.25–fce	120	30

Electrode core: smw—solid metal wire; fce—flux-cored electrode.

**Table 4 materials-13-02152-t004:** Chemical composition of clear WM performed with four different qualities of electrodes.

No.	Chemical Composition of Clear WM, wt %											
	C	Si	Mn	Cu	Mo	Ni	Cr	S	P	V	Ti	Nb
1.1	0.04	0.32	0.95	0.13	0.31	1.04	0.09	0.010	0.013	/	/	/
1.2	0.09	0.59	1.34	0.08	0.59	2.85	0.073	0.02	0.023	/	/	/
2.1	0.015	0.111	0.68	0.067	0.241	1.868	/	/	/	<0.003	<0.003	<0.003
2.2	0.026	0.546	0.89	0.090	0.307	3.245	/	/	/	0.014	0.009	<0.003

**Table 5 materials-13-02152-t005:** Type of the applied electrode, the number of the welded sample, heat input, and WM mechanical properties.

No.	Electrode Designation	Mechanical Properties of WM				Welding Joint			
		*R_e_*, MPa	*R_m_*, MPa	*A*_5_, %	KV, J (−40 °C)	*R_m_*, MPa	Fracture Location	Bending Angle α, °	
								Weld Face	Weld Root
1.1.	IHIS 1.1Ni Mo B	510	580–710	22	47	510	BM	180	140
1.2.	IHIS 2.5Ni Mo B	550–640	650–750	22–26	60–90	550–640	BM	180	180
2.1.	IHIS 1.1Ni Mo B-pp	510	580–710	22	47	510	BM	180	180
2.2.	IHIS 3.3Ni Mo B-pp	590	650–750	20	47	590	BM	180	180

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
