# Peer review of "Development of Coated Electrodes with Solid Wire and Flux-Cored Alloyed Wire for Microalloyed Steel Welding"

_materials, 2020, doi:10.3390/ma13092152_

Round 1
Reviewer 1 Report
I understood that the paper evaluated the low temperature mechanical properties of the weld metal and HAZ of J55 welded with different electrode rods.
- The amount of Ni and Mo in the molten metal increases due to different electrode rods, but the reason is not explained. Is this the effect of electrode rod composition? In that case, it is necessary to describe the components of the electrode rod.
- For the molten metal part and the HAZ part, indicate not only the component analysis but also the crystal grain size, orientation, and what kind of crystal is formed using EBSD.
- Fractures are shown, but do not only look at the microscopic area, but make a general observation.
- I can't understand Figure 7 because of the lack of focus and the lack of brightness.
- The scale of Figure 8 is too small to read.
Author Response
- In the paper, the added text is explaining the approach to experimental work and the acquisition of two new qualities of the coated electrodes with solid wire and flux-cored alloyed wire intended for welding microalloyed steels by standards and known manufacturers of additional materials. The variations of alloying elements: Mn, Ni and Mo are additionally explained. The development of filled wires with a greater thickness of metal sheath and active filling gives a significant improvement, which is the reason for replacing the solid wire with flux-cored wire, especially when it comes to alloying weld metal from the core or electrode coating.
- The paper has been substantially corrected, adjusted and modified according to your comments. Our goal was to be as clear and concrete as possible with the selected test results. The authors tried to present selected and validated results in the paper that give a clear whole in terms of content.
- Part of the text and observations have been modified and incorporated into the existing text according to your suggestions.
- and 5) Based on your comments and understandings, Figures 7-9 have been partially modified and new figures of microstructures have been added, Figures 7-13.
Reviewer 2 Report
Line 25: Please list down the benefits of the flux-core and solid alloyed wire developed in this study with conventional solid microalloyed steel wires.
Line 59: The author mentions that proportion of AF depends on chemical composition of the WM and the cooling rate. Has the cooling rate been kept constant in all the experiments in this study ?
Table 3: The selection windows are very narrow for Mo and Ni %.
Also, why was the % of one of the elements (Ni, Mo) varied while keeping the other fixed ?
Line 121: How was impact energy measured on the samples ?
Table 5: Measurements for specimen E1 have been shown in Figures 3,4,5. However their properties are not listed in Table 5.
Author Response
Line 25: According to your suggestions, a text has been added. It is explaining the advantage of a core of the flux-cored wire compared to a core of the classic solid wire.
Line 59: The textual explanation has been substantially changed and added. Based on the results of preliminary research, welding parameters have been selected that allow a very low impact energy input of 7.5-8.5 kJ / cm, with a temperature limitation between two welds at max 200°C and an adequate cooling rate in the temperature range of 800-500°C. For experimental welding, the panels were of the same batch, selected from 7mm thick J5 microalloyed steel.
Table 3: Due to additional clarifications, the textual part was significantly enlarged, including the number of tables and figures showing the microstructures. So Table 3 is now the new Table 2 to which another electrode has been added (total applied in the experiment: two with solid classical wire core and two with the flux-cored wire). Mn and Mo are slightly altered but Ni is altered at two levels at both types of electrode. The solid wire core electrodes has the following weld metal composition: No.1.1 (0.95% Mn + 1.04% Ni + 0.31% Mo) and No.1.2 (1.34% Mn + 2.850% Ni + 0 , 59% Mo). The flux-cored wire electrodes has the following weld metal composition: No.2.1 (0.68% Mn + 1.868% Ni + 0.241% Mo) and No.2.2 (0.89% Mn + 3.245% Ni + 0.307% Mo).
Line 121: The impact energy is measured on the samples using an instrumented Charpy pendulum. The analysis of weladed joints performed with low heat input, the thickness of base material and with temperature limitation between two passages at max 200 oC and appropriate cooling rate, taking into account the ambient temperature, was chosen.
Table 5: Your observations led us to limit the mentioned part of the study to the optimum low level of heat input (impact energy: E = 7.5-8.5 kJ/cm) when welding microalloyed steels. Toughness at test temperatures is only given for the low intake of welding heat due to the reduced volume of the work itself. All of these changes, adjustments, and additions have further affected the chain and even the amendment of the text of the conclusions themselves.
Reviewer 3 Report
- In lines 123-131 on page 4, the authors explain the effect of Mo and Ni on the increase of tensile strength and yield strength. However no mention is made of the increase of Mn (and Cr). Does it have a role as well ?
- In figures 8 a and b, none of the micrographs demonstrate any polygonal ferrites. Are only acicular and proeutectoid ferrites present ?
- Is increased Ni solely responsible for increase in toughness (by the increase of acicular ferrite), or do other ingredients also play a role ? It would be interesting to know how the other elements (Mo, Mn, Cr) also affect this property.
Author Response
- The development and production of additional materials is a very complex and challenging research task so that in addition to the role of the electrode lining, alloying elements in pure weld metal have a significant role in the properties of welds. In the paper, a textual analysis of the alloying elements Mn and Mo was added. This was slightly altered in the composition of the electrodes. Ni was altered at two levels in both types of electrodes. Additionally, as you suggested, the role of alloying elements whose total cumulative amount in the weld metal is: 2.92% (sample No.1.1); 5.66% (sample No.1.2); 2.99% (sample No.2.1) and 5.13% (sample No.2.2). The proportion of Cr in the weld metal was not tested due to the choice of the specified electrodes type (without chromium content) used for research work.
- Based on your comments, better quality photographs were selected and the number of new microstructure photographs increased (Figures 8-13). In the weld metal of the welds shown, the presence of different ferrite morphological forms was observed. Acyclic ferrite (AF), polygonal ferrite (PF) and secondary phase ferrite (FS) are most prevalent.
- Based on your suggestions, a broader explanation of the influence of alloying elements and other factors on the toughness of welded joints microalloyed steels at different test temperatures is given in the paper. Optimization of the structure of the weld metal (dominant presence of acicular ferrite) and the heat-affected zone (the narrow overheated zone where carbide, nitride and carbonitride precipitates dissolve and grain growth) is achieved by limited heat input (for E-welds at level 5-10 kJ/cm) by limiting the temperature between the two welds at max 200 oC and the corresponding cooling rate in the temperature interval 800 – 500 oC.
Reviewer 4 Report
The authors studied the quality of a new rutile electrode by testing their chemical composition and mechanical properties. The topic is worth of investigation and the results are relevant for the field. However, the manuscript is not well prepared making it very hard to follow. The English needs an extensive revision and significant improvements throughout the text. Some problems were given as below:
- What's the mean "qualitative analysis of the WM using SEM" (Line 103)? Chemical composition or Size?
- What's the purpose of etching of microstructure ? Please explain. (Line 104)
- Please explain why the quantitative analysis can confirm the higher proportion of acicular ferrite? Please provide more evidence about the formation of a higher proportion of AF. (Line 126-Line 127)
- The phrase "influenced the fragmentation of the primary structure" is confused, the tendency is increasing or decreasing ? Please clarify (Line 129)
- Please explain the mean of the code of specimen marking in Table 5.
- Figure 2b in Line 169?
- How did the authors get the volume fraction of acicular ferrite ? (Line186, 187)
- Did the label "PF" in the Figure 8 represent "proeutectoid ferrite"or "polygonal ferrite"? Please explain clear.
- The Conclusions are weak, being mainly a repeat of the description of the results. Try numbering the Conclusions - it helps focus the mind. Write those matters that you want the reader to have learned from your work.
- Some complicated sentences should be replaced by some shorter and understandable sentences.
- The sentence in Line 67-Line 69 should be placed in "Materials and Methods".
- Please provide the product model, manufacturer, city and state of the manufacturer of the materials and equipment.
Author Response
- Line 103: "qualitative analysis of WM using SEM" means a chemical composition.
- Line 104: In our opinion, the above part of the text is unnecessary and has been deleted.
- Lines 126 and 127: Your comments indicated the need to correct and supplement the test results according to the set goal, which is why the text related to the results in Table 5 was deleted and replaced with other results and comments. The comments led to a significant change to the text related to the explanation of the cause of the increase in the acyclic ferrite (AC) content in the structure of the weld metal by designing good quality new coated electrodes intended for welding microalloyed steels.
- Line 129: Text in this line is completely changed too as the previous text.
- Table 5. The previous text is directly related to the data in Table 5, which is why it was completely deleted and modified based on the comments of the reviewers, and therefore the comment of the statistical processing of the mechanical properties values in the function of changing the welding heat intake is useless.
- Line 169: The contents of Table 2 have been completely amended.
- Lines 186 and 189: The percentage of austenite in the structure was done on an optical microscope by subjective evaluation.
- The presence of various ferrite morphological forms was observed in the weld metal of the welds shown. The most obscured are acicular ferrite (AF), proeutectoid ferrite (PF) and ferrite with secondary phase (FS).
- All previous changes and selection of research results led to a change of concept and a complete change of conclusions.
- A suggestion regarding the length of sentences has been adopted.
- Lines 67-69: Recommendations are accepted.
Round 2
Reviewer 4 Report
The quality of the article has been significantly improved. Only some minor error: no scale labels in Figure 7, 8, 9, 10, 11, Figure 12 a. and b., Figure 13 a. and b., Please check.
I recommend publication in Materials.